# Incentivizing Dual Process Thinking for Efficient Large Language Model Reasoning

**Xiaoxue Cheng**[1,2,3*], **Junyi Li**[4*], **Zhenduo Zhang**[5], **Xinyu Tang**[1],
**Wayne Xin Zhao**[1,2,3†], **Xinyu Kong**[5], **Zhiqiang Zhang**[5]
[1] Gaoling School of Artificial Intelligence, Renmin University of China
[2] Beijing Key Laboratory of Research on Large Models and Intelligent Governance
[3] Engineering Research Center of Next-Generation Intelligent Search and Recommendation, MOE
[4] Department of Data Science, City University of Hong Kong [5] Ant Group
chengxiaoxue@ruc.edu.cn, junyili@cityu.edu.hk, batmanfly@gmail.com

## Abstract

Large reasoning models (LRMs) have demonstrated strong performance on complex reasoning tasks, but often suffer from overthinking, generating redundant content regardless of task difficulty. Inspired by the dual process theory in cognitive science, we propose **A**daptive **C**ognition **P**olicy **O**ptimization (**ACPO**), a reinforcement learning framework that enables LRMs to achieve efficient reasoning through adaptive cognitive allocation and dynamic system switch. ACPO incorporates two key components: (1) introducing system-aware reasoning tokens to explicitly represent the thinking modes thereby making the model's cognitive process transparent, and (2) integrating online difficulty estimation and token length budget to guide adaptive system switch and reasoning during reinforcement learning. To this end, we propose a two-stage training strategy. The first stage begins with supervised fine-tuning to cold start the model, enabling it to generate reasoning paths with explicit thinking modes. In the second stage, we apply ACPO to further enhance adaptive system switch for difficulty-aware reasoning. Experimental results demonstrate that ACPO effectively reduces redundant reasoning while adaptively adjusting cognitive allocation based on task complexity, achieving efficient hybrid reasoning.

## 1 Introduction

Recent advances in large reasoning models (LRMs) [1] have demonstrated remarkable success on complex tasks such as mathematical reasoning [2, 3, 4, 5], largely attributed to reinforcement learning that encourages the generation of detailed, step-by-step reasoning processes. LRMs improve answer accuracy through self-reflection and self-verification during long reasoning paths. As the reasoning length increases, the performance of model tends to improve accordingly [2, 6, 7]. Although the long chain-of-thought (CoT) [8] reasoning in LRMs is effective for solving complex problems [2, 9], it often leads to *overthinking* [10, 11], producing redundant reasoning paths. Most existing LRMs rely on fixed reasoning strategies, lacking the ability to dynamically switch between different thinking modes based on task complexity. This rigidity results in inefficient inference, particularly for simple problems that could be resolved more effectively with concise and direct reasoning.

Several recent efforts have explored long CoT compression for efficient reasoning [12, 13]. One line of work fine-tunes LRMs using supervision from shorter chain-of-thought exemplars [14, 15], encouraging the model to arrive at correct answers with fewer intermediate steps. Another line

---

*Equal Contribution.
†Corresponding author.

39th Conference on Neural Information Processing Systems (NeurIPS 2025).

introduces length penalties into reinforcement learning reward functions [3, 16, 17, 18], explicitly discouraging unnecessarily long reasoning trajectories during training. Since many prior approaches overlook task difficulty and treat all samples uniformly with the sole objective of shortening reasoning paths, some recent methods attempt to address this by estimating length budgets offline and training the model on sampled trajectories accordingly [19]. However, such offline strategies depend on precomputed budgets and fixed preference data, restricting their adaptability and scalability.

Inspired by the dual process theory in cognitive science [20], which states that humans have two systems for thinking — fast, intuitive thinking (System 1) and slow, deliberate thinking (System 2), we pose the following research question: *Can LRMs learn to dynamically switch between fast and slow thinking modes based on task complexity, enabling more efficient and adaptive reasoning?*

In this paper, we propose **A**daptive **C**ognition **P**olicy **O**ptimization (**ACPO**), a reinforcement learning framework that enables LRMs to perform efficient and adaptive reasoning through dynamic system switch between fast and slow thinking modes. To achieve this, we first introduce system-aware reasoning tokens (*e.g.,* `<fast_think>`, `<slow_think>`) to explicitly indicate the model engagement in fast or slow thinking modes during the reasoning process. Based on these tokens, we construct a dataset from which the reasoning trajectory is interleaved with fast thinking and slow thinking steps, followed by the final answer. Then, we propose a two-stage training strategy to enable dynamic system switch reasoning. In the first stage, we perform supervised fine-tuning with the constructed dataset as a cold start, establishing the model's foundational ability to generate explicit thinking process. In the second stage, we apply reinforcement learning with ACPO for further enhancement. Specifically, we introduce an online token length budget (TLB) mechanism that estimates task difficulty based on the model's sampling success rate, providing a real-time signal to adjust the reasoning budget according to task complexity. These estimates are further incorporated into a reward function that guides cognitive allocation through two components: a TLB reward that encourages difficulty-aware length control, and a system pattern reward that incentivizes appropriate system switch between fast and slow thinking modes.

We evaluate ACPO on a range of complex reasoning benchmarks. Unlike traditional reinforcement learning methods that focus solely on accuracy or fixed-length compression, ACPO dynamically adjusts both reasoning length and cognitive effort based on task difficulty. For challenging problems, it effectively reduces redundant reasoning steps, while for simpler tasks, it avoids overcompression and maintains high accuracy. These results highlight the advantages of difficulty-aware reasoning, demonstrating the effectiveness of dynamic cognitive control through adaptive reward optimization. The main contributions of this work are as follows:

- We introduce system-aware reasoning tokens to explicitly annotate fast and slow thinking steps in LRMs, enabling transparent reasoning paths and providing a foundation for dynamic control of the model's cognitive strategies.

- We propose ACPO, a reinforcement learning framework that integrates online difficulty estimation and token length budget to dynamically calibrate the reward function and steer difficulty-aware cognitive allocation.

- We demonstrate through extensive experiments that ACPO effectively compresses redundant reasoning content while maintaining accuracy, achieving a robust balance between efficiency and performance by avoiding overcompression and underexploration.

## 2   Preliminary

### 2.1   Group Relative Policy Optimization (GRPO)

Group Relative Policy Optimization (GRPO) [21] is a reinforcement learning framework that eliminates the need for a value function by estimating advantages in a group-relative manner. Given a specific question-answer pair $(q, a)$, the behavior policy $\pi_{\theta_{\text{old}}}$ samples a group of $G$ individual responses $\{y_i\}_{i=1}^{G}$. For each token $y_{i,t}$ in response $y_i$, its normalized advantage is computed based on the group-level rewards $\{R_i\}_{i=1}^{G}$ as follows:

$$\hat{A}_{i,t} = \frac{R_i - \text{mean}(\{R_i\}_{i=1}^{G})}{\text{std}(\{R_i\}_{i=1}^{G})}. \tag{1}$$

Similar to Proximal Policy Optimization (PPO) [22], GRPO adopts a clipped surrogate objective with an additional KL regularization term to stabilize optimization:

$$\mathcal{J}_{\text{GRPO}}(\theta) = \mathbb{E}_{(q,a)\sim\mathcal{D},\{y_i\}_{i=1}^G\sim\pi_{\theta_{\text{old}}}(\cdot|q)}$$

$$\left[ \frac{1}{G}\sum_{i=1}^{G}\frac{1}{|y_i|}\sum_{t=1}^{|y_i|}\left( \min\left( r_{i,t}(\theta)\hat{A}_{i,t}, \ \text{clip}\left( r_{i,t}(\theta), 1-\varepsilon, 1+\varepsilon \right)\hat{A}_{i,t} \right) - \beta D_{\text{KL}}(\pi_\theta||\pi_{\text{ref}}) \right) \right], \quad (2)$$

where $r_{i,t}(\theta) = \frac{\pi_\theta(y_{i,t}|q,y_{i,<t})}{\pi_{\theta_{\text{old}}}(y_{i,t}|q,y_{i,<t})}$ is the importance sampling ratio for the $t$-th token $y_{i,t}$.

## 2.2 Token Length Budget

The token length budget (TLB) [19] is a method introduced for estimating an appropriate token-level budget for reasoning trajectories. Instead of prescribing a fixed or hard constraint, TLB adaptively adjusts the expected output length based on the sampling accuracy of candidate responses, capturing the inherent difficulty of the task. Specifically, for a given question $q$, $N$ candidate responses are sampled from a language model. The TLB, denoted as $L_{\text{budget}}$, is computed as:

$$L_{\text{budget}} = p \cdot L_r + (1-p) \cdot L_{\max}, \quad (3)$$

where $p = \frac{c}{N}$ represents the sampling success rate, with $c$ denoting the number of correct responses, $L_r$ denotes the average token length among the correct responses, $L_{\max}$ is the maximum token length among all responses. This formulation allows the budget to adapt to task complexity, providing flexible length guidance based on sampling accuracy without external supervision. In the DAST method [19], TLB is computed in an offline manner by sampling multiple responses for each question before training. These responses are then converted to preference pairs for further fine-tuning according to their estimated TLB scores.

# 3 Method

In this section, we propose **ACPO**, a reinforcement learning framework that enables dynamic and adaptive system switch of LRMs to improve reasoning efficiency and adaptability to task complexity. This framework incorporates system-aware reasoning tokens to explicitly represent fast and slow thinking steps of LRMs, and integrates an online length budget estimation in RL training to dynamically adjust reasoning length based on task difficulty, guiding the model to balance accuracy and efficiency through adaptive cognitive effort allocation. We first describe the use of system-aware reasoning tokens for system explicitization, followed by the construction of explicit reasoning paths for supervised fine-tuning. Next, we present the reinforcement learning process of ACPO, providing a detailed overview of the online token length budget estimation and reward design. The overall framework of the proposed ACPO is illustrated in Figure 1.

## 3.1 Explicit Dual Process Reasoning

The dual process theory [20] models human thinking as a combination of fast, intuitive processes (System 1) and slower, deliberate reasoning (System 2). Inspired by this theory, we introduce *system-aware reasoning tokens* as explicit indicators of different thinking modes within the model's reasoning process. Specifically, we define four special tokens, `<fast_think>`, `</fast_think>`, `<slow_think>` and `</slow_think>`, to explicitly wrap fast and slow reasoning steps, respectively. The introduction of reasoning tokens serves two purposes. First, it makes the model's internal thinking modes explicitly observable, revealing the process of system switch. Second, it provides a fine-grained mechanism for controlling and monitoring the model's cognitive dynamics. With this explicit thinking process, we can train the model for system switch, optimizing its cognitive allocation based on task difficulty.

### 3.1.1 Data Construction

To enable the model to learn explicit thinking patterns, we construct a training dataset for supervised fine-tuning. Each sample consists of a question paired with its corresponding answer consisting of interleaved fast thinking and slow thinking segments. To ensure the quality of the training data, we utilize the LIMO dataset [23], a carefully curated, high-quality dataset of complex mathematical

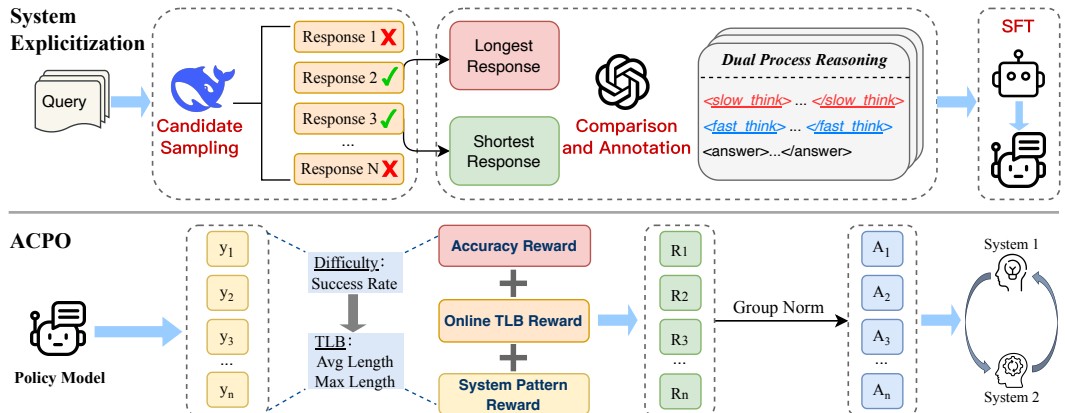

Figure 1: The overall framework of ACPO. The upper section illustrates the system explicitization process and cold start training via SFT. The lower section presents the ACPO training phase.

reasoning tasks with high challenge and diverse knowledge coverage. These tasks demand precise, multi-step reasoning to arrive at correct answers, making them well-suited for constructing explicit thinking processes. Specifically, our data construction involves two main steps: *Candidate Response Sampling* and *Response Comparison and Annotation*.

• **Candidate Response Sampling:** Building on the approach in TOPS [24], we prompt the *DeepSeek-R1-Distill-Qwen-32B* [2] model to generate multiple candidate responses with varying lengths for each question. We adopt the prompting strategies in TOPS to encourage diversity of the reasoning lengths. After generating these candidates, we filter out incorrect responses and select the longest and shortest correct answers for further processing.

• **Response Comparison and Annotation:** We employ *GPT-4* [25] as an evaluator to perform fine-grained comparison and annotation between the selected response pairs. For those reasoning steps that are present in both short and long responses, we consider them as essential and detailed components that should be marked as slow thinking, as they represent critical, non-omittable reasoning processes. For other steps that appear in the long response but are omitted or summarized in the short response, we view them as trivial steps that should be labeled as fast thinking. Based on this principle, we reframe the short response by enclosing essential reasoning steps with <slow_think></slow_think> tags and trivial reasoning steps with <fast_think></fast_think> tags. An illustrative example is provided in Figure 4.

### 3.1.2 Supervised Fine-Tuning (Cold Start)

After completing the data construction process, we obtain a dataset containing 745 annotated samples with explicit reasoning tokens. This dataset is used for supervised fine-tuning, allowing the model to learn to generate outputs with clearly interleaved fast and slow thinking modes. This cold start phase equips the model with the foundational ability to explicitly switch between different thinking modes. We train the model with a standard cross-entropy loss to ensure its thinking process aligns with the intended system switch patterns.

### 3.2 Reinforcement Learning with ACPO

While the SFT process allows the model to align its outputs with the annotated reasoning tokens, explicitly capturing fast and slow thinking patterns, this approach may lead to overfitting and a tendency to memorize fixed reasoning paths. To overcome this limitation, we introduce a reinforcement learning phase, optimizing the fine-tuned model to achieve more flexible and adaptive cognition allocation through interaction and exploration. In the following sections, we present the proposed ACPO method, including the online token length budget estimation and reward design that are specifically tailored for dynamic system switch.

### 3.2.1 Reward Design

In this work, we adopt GRPO as the underlying RL algorithm, extending its framework with a customized reward structure to explicitly guide the model's reasoning behavior. Formally, during the ACPO optimization process, we sample a set of $N$ candidate responses $\{y_1, y_2, \ldots, y_N\}$ from the current policy model for each training query $q_i$. For each response, we compute a composite reward that consists of three components: accuracy reward, online TLB reward, and system pattern reward to encourage accurate and efficient reasoning with adaptive cognition allocation, as described below.

**Accuracy Reward.** For each sampled response $y_i$, we assess its correctness by comparing the generated answer with the ground-truth label in the training data. The accuracy reward is assigned as:

$$R_{\text{acc},i} = \begin{cases} +1, & \text{if } y_i \text{ is correct,} \\ -1, & \text{otherwise.} \end{cases} \tag{4}$$

**Online TLB Reward.** To guide the model in generating reasoning paths of appropriate length that align with the task difficulty, we introduce an online token length budget (TLB) reward. Unlike the offline approach in DAST, which requires extensive pre-sampling and pairwise preference construction, our method estimates the length budget $L_{\text{budget}}$ online during training, without additional computational overhead. This is achieved by leveraging the natural group-based sampling process in GRPO, allowing the sampling success rate $p$ and token length budget $L_{\text{budget}}$ to be directly calculated from the existing candidate responses, as defined in Eq. 3. Given a sampled response $y_i$ with actual length $L_i$, we compute its TLB reward based on the deviation from the estimated $L_{\text{budget}}$. The online TLB score $R_{\text{TLB},i}$ for each response is then defined as:

$$R_{\text{TLB},i} = \begin{cases} \tanh(-\lambda_i), & \text{if } y_i \text{ is correct,} \\ \tanh(\lambda_i), & \text{otherwise,} \end{cases} \tag{5}$$

where $\lambda_i = \frac{L_i - L_{\text{budget}}}{L_{\text{budget}}}$ and the $\tanh$ function is used to smoothly bound the reward within (-1, 1). The TLB reward adapts the model's reasoning strategy to the difficulty of each task, promoting efficient fast thinking for easier problems while allowing longer and more deliberate reasoning for harder problems. Additionally, the online TLB estimation enables real-time adaptation and avoids reliance on static preference data, leading to better generalization across diverse tasks.

**System Pattern Reward.** To further encourage difficulty-aware system switch, we use the sampling success rate $p$ from the TLB estimation as a proxy for task difficulty. For easier queries with higher $p$, the model is encouraged to allocate a larger proportion of its reasoning path to fast thinking. In contrast, for harder queries with lower $p$, the model is incentivized to allocate more steps to slow thinking for more careful deliberation. Given a sampled response $y_i$, we compute the proportions of fast and slow thinking tokens within the total reasoning sequence, denoted as $\rho_{\text{fast},i}$ and $\rho_{\text{slow},i}$, respectively. The system pattern reward is then defined as:

$$R_{\text{think},i} = \begin{cases} \rho_{\text{fast},i}, & \text{if } p > p_{\text{thresh}}, \\ \rho_{\text{slow},i}, & \text{otherwise,} \end{cases} \tag{6}$$

where $p_{\text{thresh}}$ is a predefined complexity threshold separating easy and hard questions.

The final reward $R_i$ for each sampled response $y_i$ is computed as a weighted combination of the three components:

$$R_i = \begin{cases} \max\left(w_{\text{acc}} \cdot R_{\text{acc},i} + w_{\text{len}} \cdot R_{\text{TLB},i} + w_{\text{think}} \cdot R_{\text{think},i},\ 0.1\right), & \text{if } y_i \text{ is correct,} \\ \min\left(w_{\text{acc}} \cdot R_{\text{acc},i} + w_{\text{len}} \cdot R_{\text{TLB},i} + w_{\text{think}} \cdot R_{\text{think},i},\ -0.1\right), & \text{if } y_i \text{ is incorrect.} \end{cases} \tag{7}$$

The weights $w_{\text{acc}}$, $w_{\text{len}}$, and $w_{\text{think}}$ are hyperparameters that balance the importance of different reward signals during optimization. In order to ensure that the reward for correct responses remains strictly positive and the reward for incorrect responses remains strictly negative, we apply $\max$ and $\min$ operations to clip the final reward within the desired range. Based on the final reward, we leverage the GRPO objective in Eq. 2 to optimize the policy model.

Table 1: Evaluation results of ACPO on three different reasoning models across the MATH 500 and AIME 2024 datasets. **Bold** fonts indicate the best performance for each reasoning model.

| Methods | MATH 500 | | | AIME 2024 | | |
|---|---|---|---|---|---|---|
| | Accuracy | #Token | ACU | Accuracy | #Token | ACU |
| DeepSeek-R1-Distill-Qwen-1.5B | **83.9** | 5708 | 0.98 | 28.9 | 16894 | 0.11 |
| THINKPRUNE | 82.9 | 2356 | 2.35 | 27.0 | 7574 | 0.24 |
| ACPO-1.5B | 81.0 | **1679** | **3.22** | **30.0** | **6670** | **0.30** |
| DeepScaleR-1.5B-Preview | 87.8 | 3914 | 1.50 | 43.1 | 17206 | 0.17 |
| L1-Exact | 79.8 | 1044 | 5.09 | 16.7 | 1798 | 0.62 |
| L1-Max | 81.8 | 999 | 5.46 | 23.3 | 2230 | 0.69 |
| DeepSeek-R1-Distill-Qwen-7B | **92.8** | 3977 | 0.33 | **55.5** | 13254 | 0.06 |
| SFT_Shortest | 91.8 | 2954 | 0.44 | 50.0 | 10757 | 0.07 |
| SimPO_Shortest | 87.8 | **970** | **1.29** | 33.3 | **2737** | **0.17** |
| SimPO_DAST | 92.6 | 2802 | 0.47 | 53.3 | 6337 | 0.12 |
| ACPO-7B | 91.6 | 1405 | 0.93 | 52.8 | 4520 | **0.17** |
| DeepSeek-R1-Distill-Llama-8B | **89.1** | 5003 | 0.22 | 42.9 | 16374 | 0.04 |
| ACPO-8B | 87.4 | **2232** | **0.49** | **43.3** | **7405** | **0.07** |

## 4 Experiment

### 4.1 Experimental Setup

**Datasets and Evaluation Metrics.** We conduct training on the DeepScaleR-Preview-Dataset [26], a mathematical dataset consisting of 40K question-answer pairs drawn from AIME, AMC, Omni-Math [27] and STILL [28]. For evaluation, we assess model performance on three mathematical datasets: GSM8K [29], AIME 2024, and MATH 500 [30]. For each test question, we generate 16 responses using a sampling temperature of $0.6$ and a top-$p$ value of $0.95$, and compute *pass@1* to measure accuracy. We report the average number of tokens generated per response in each dataset to assess reasoning efficiency. Moreover, we use Accuracy per Computation Unit (ACU) metric [31] to capture the balance between reasoning accuracy and efficiency, which is defined as :

$$\text{ACU} = \frac{\text{Accuracy}}{\text{\#Params} \times \text{\#Tokens}} \tag{8}$$

**Baselines.** We conduct experiments on DeepSeek-R1-Distill-Qwen-1.5B, DeepSeek-R1-Distill-Qwen-7B, and DeepSeek-R1-Distill-Llama-8B [2] and compare ACPO against the following methods.

- **DeepScaleR-1.5B-Preview** [26] is a model trained on DeepSeek-R1-Distill-Qwen-1.5B with GRPO. We include it as an important baseline in our evaluations.

- **THINKPRUNE** [17] trains models with a fixed token limit, pruning unfinished responses beyond this limit with zero reward. We include *THINKPRUNE* with 2k token length constraint trained on DeepSeek-R1-Distill-Qwen-1.5B for comparison.

- **L1** [18] trains models to follow specified response lengths by introducing exact and maximum length penalties in reinforcement learning, yielding two variants, *L1-Exact* and *L1-Max*, both trained on *DeepScaleR-1.5B-Preview*.

- **DAST** [19] introduces an offline approach to estimate token length budgets through sampling and construct length preference data for SimPO [32] training. We include the comparison methods *SFT_Shortest*, *SimPO_Shortest*, and *SimPO_DAST* from DAST in our experiments.

**Implementation Details.** In the cold start phase, we fine-tune the models for 3 epochs using 745 annotated samples with explicit reasoning tokens. For ACPO training, we adopt the same hyperparameter settings as used in DeepScaleR-1.5B-Preview. Specifically, we use a learning rate of $1 \times 10^{-6}$, a batch size of 128, and a maximum context length of 8K tokens during training. The models are trained for one epoch, and both the SFT and RL stages are conducted using the VeRL

Table 3: Performance comparison between three reasoning models trained with GRPO and ACPO on the MATH 500 and AIME 2024 datasets.

| Methods | MATH 500 | | | AIME 2024 | | |
|---|---|---|---|---|---|---|
| | Accuracy | #Token | ACU | Accuracy | #Token | ACU |
| DeepSeek-R1-Distill-Qwen-1.5B | 83.9 | 5708 | 0.98 | 28.9 | 16894 | 0.11 |
| +GRPO | 84.5 | 3098 | 1.82 | 29.0 | 12990 | 0.15 |
| +ACPO | 81.0 | 1679 | 3.22 | 30.0 | 6670 | 0.30 |
| DeepSeek-R1-Distill-Qwen-7B | 92.8 | 3977 | 0.33 | 55.5 | 13254 | 0.06 |
| +GRPO | 92.5 | 3700 | 0.36 | 53.2 | 8577 | 0.08 |
| +ACPO | 91.6 | 1405 | 0.93 | 52.8 | 4520 | 0.17 |
| DeepSeek-R1-Distill-Llama-8B | 89.1 | 5003 | 0.22 | 42.9 | 16374 | 0.04 |
| +GRPO | 90.4 | 2172 | 0.52 | 43.3 | 8883 | 0.06 |
| +ACPO | 87.4 | 2232 | 0.49 | 43.3 | 7405 | 0.07 |

framework [33]. We set $p_{\text{thresh}} = 0.5$ in Eq. 6 and set the reward weights as $w_{\text{acc}} = 0.6$, $w_{\text{len}} = 0.3$, and $w_{\text{think}} = 0.1$ in Eq. 7.

## 4.2 Main Results

The evaluation results of our method and the baselines are presented in Table 1 and Table 2.

For more challenging datasets, such as MATH 500 and AIME 2024 in Table 1, ACPO achieves significant token reduction while maintaining competitive accuracy. For instance, on the AIME 2024 dataset, ACPO-1.5B reaches 30.0% accuracy with an average token count of 6670, representing a 60.5% reduction in token usage compared to 16,894 tokens required by DeepSeek-R1-Distill-Qwen-1.5B.

For MATH 500 dataset, ACPO reduces the token count for DeepSeek-R1-Distill-Qwen-7B from 3977 to 1405 with only a slight accuracy decrease. Although L1-Max and L1-Exact also achieve substantial token compression, they suffer noticeable accuracy drops compared to DeepScaleR-1.5B-Preview. This demonstrates that ACPO can effectively shorten reasoning lengths without sacrificing too much accuracy, particularly on complex reasoning tasks.

For simpler GSM8K dataset in Table 2, where the reasoning paths of DeepSeek-R1-Distill-Qwen-1.5B and 7B tend to be less redundant,

Table 2: Evaluation results of ACPO with the three reasoning models on the GSM8K dataset.

| Methods | Accuracy | #Token | ACU |
|---|---|---|---|
| R1-Distill-Qwen-1.5B | 79.9 | 643 | 8.28 |
| ACPO-1.5B | 81.3 | 572 | 9.48 |
| R1-Distill-Qwen-7B | 86.5 | 445 | 2.78 |
| ACPO-7B | 88.3 | 413 | 3.05 |
| R1-Distill-Llama-8B | 82.9 | 1026 | 1.01 |
| ACPO-8B | 86.7 | 732 | 1.48 |

ACPO maintains the original length scale while achieving notable accuracy improvements. Specifically, ACPO-1.5B reduces average token usage from 643 to 572 while improving accuracy from 79.9% to 81.3%. Similarly, ACPO-7B achieves 88.3% accuracy with 413 tokens, outperforming the baseline in both accuracy and efficiency. However, for DeepSeek-R1-Distill-Llama-8B, which has relatively long reasoning paths with 1026 tokens, ACPO achieves effective compression by reducing the token count to 732 tokens with 3.8% accuracy improvement. These results highlight the adaptability of the online token length budget estimation, enabling selective compression that preserves necessary reasoning length while avoiding excessive compression.

## 4.3 Further Analysis

### 4.3.1 Ablation Analysis

To assess the impact of the reward design in ACPO on both accuracy and reasoning length, we conduct ablation studies by training models with the GRPO reward setting, keeping all other parameters identical to those in ACPO. Specifically, we evaluate the three models on the MATH 500 and AIME 2024 datasets: DeepSeek-R1-Distill-Qwen-1.5B, DeepSeek-R1-Distill-Qwen-7B, and DeepSeek-R1-

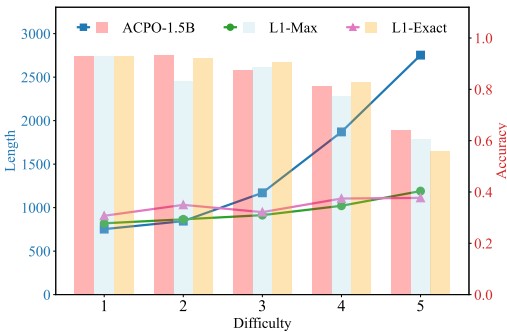
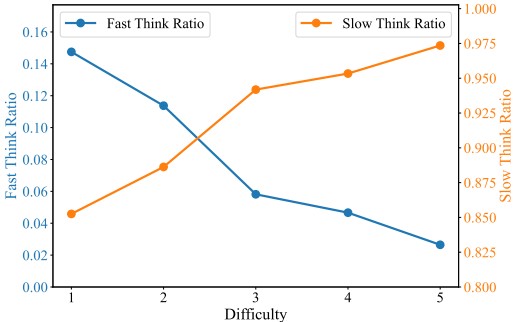

Figure 2: Average response length and accuracy across different difficulty levels on MATH 500.

Figure 3: Average fast and slow thinking ratios across different difficulty levels on MATH 500.

Distill-Llama-8B, each trained for one epoch with GRPO and ACPO. We report accuracy, average token length, and ACU scores for comparison in Table 3.

From the results, we observe that both GRPO and ACPO can compress the reasoning length of the original models. In contrast, ACPO consistently achieves more significant compression, often exceeding 50% reduction in token length. For instance, it reduces the average reasoning length of DeepSeek-R1-Distill-Qwen-1.5B on the MATH dataset to 1679 tokens, compared to 3098 tokens for GRPO. In terms of accuracy, GRPO shows a slight advantage, especially on the MATH dataset. However, ACPO demonstrates comparable performance on the more challenging AIME dataset. For DeepSeek-R1-Distill-Qwen-1.5B, ACPO reaches an accuracy of 30.0% compared to 29.0% for GRPO, effectively balancing accuracy and efficiency. While GRPO focuses solely on correctness, the reward design of ACPO jointly optimizes answer accuracy, reasoning length, and system switch, enabling a flexible coordination between correctness, efficiency, and cognitive allocation.

### 4.3.2 Difficulty Adaptability Analysis

To validate the difficulty adaptability of ACPO, we perform analyses on the MATH 500 dataset, which is divided into five difficulty levels. In Figure 2, we compare the accuracy and average response length of ACPO-1.5B, L1-Exact, and L1-Max. Unlike the L1 approach, which applies uniform token constraints regardless of the difficulty of questions, ACPO demonstrates a more adaptive length control strategy. It effectively shortens responses for simpler problems while preserving the necessary reasoning length for complex questions, resulting in minimal accuracy loss on more challenging levels (*e.g.,* level 4, level 5). This adaptive capability highlights the advantage of our approach in difficulty-aware reasoning, aligning the token length budget more closely with problem complexity.

In Figure 3, we further analyze the fast and slow thinking ratios for ACPO-7B on the MATH dataset. It is observed that, as the problem difficulty increases, the proportion of fast thinking decreases while the slow thinking component increases. Notably, the proportion of slow thinking increases rapidly from difficulty level 1 to level 3, and then grows more gradually beyond level 3. This trend suggests that the model is able to allocate more reasoning effort when solving more difficult problems, dynamically allocating cognitive resources based on task complexity. Such adaptive behavior aligns with the dual process theory of human thinking, demonstrating the effectiveness of our method in reasonable cognition allocation and dynamic system switch.

### 4.3.3 Case Study

In Figure 4, we present an example comparing the reasoning behavior of DeepSeek-R1-Distill-Qwen-1.5B trained with GRPO and ACPO on a number theory problem from the Math 500 dataset. The task requires finding the smallest positive integer multiple of 30 that can be written using only the digits 0 and 2, which requires applying divisibility rules: the number must end in 0 and the sum of its digits must be divisible by 3. The model trained with GRPO adopts a cautious and exhaustive strategy, enumerating multiple candidate numbers (*e.g.,* 30, 60, 90) step by step, even after identifying a valid solution. This leads to a lengthy reasoning path with 1555 tokens, filled with redundant self-verification and repeated constraint checking. In contrast, the model trained with ACPO efficiently identifies the necessary constraints, rapidly filtering out invalid candidates and

**What is the least positive integer multiple of 30 that can be written with only the digits 0 and 2?**

| DeepSeek-R1-Distill-Qwen-1.5B with ACPO | DeepSeek-R1-Distill-Qwen-1.5B with GRPO |
|---|---|
| <slow_think>Okay, so I need to find the smallest positive integer multiple of 30 that can be written using only the digits 0 and 2. Hmm, let's break this down. First, I know that 30 is 3 times 10, so any multiple of 30 must be divisible by both 3 and 10. That means the number has to end with a 0 because of the divisibility by 10. Next, the number must also be divisible by 3. The rule for divisibility by 3 is that the sum of the digits must be a multiple of 3. Since the number can only have digits 0 and 2, I need to figure out how many 2s I need so that their sum is divisible by 3. Let me try with the smallest number of digits. Starting with one digit: 2. But 2 isn't divisible by 3, so that's out. Next, two digits: 20. The sum of the digits is 2 + 0 = 2, which isn't divisible by 3. So 20 doesn't work. How about three digits? Let's try 220. The sum is 2 + 2 + 0 = 4, still not divisible by 3. Next, 2220. The sum is 2 + 2 + 2 + 0 = 6, which is divisible by 3. So 2220 should work.</slow_think> <fast_think>Wait, let me double-check. 2220 divided by 30 is 74, which is an integer. So yes, 2220 is a multiple of 30 and only uses the digits 2 and 0. I don't think there's a smaller number that fits these criteria because the previous attempts with fewer digits didn't work.</fast_think> The least positive integer multiple of 30 that can be written with only the digits 0 and 2 is 2220.<answer>The least positive integer multiple of 30 that can be written with only the digits 0 and 2 is \boxed{2220}.</answer> **(476 Tokens)** | <think>Okay, so I need to find the smallest positive integer multiple of 30 that can be written using only the digits 0 and 2. Hmm, let's break this down. First, I know that 30 is equal to 3 multiplied by 10, so any multiple of 30 must be a multiple of both 3 and 10. That means the number I'm looking for has to end with a 0 because of the 10 part, and the sum of its digits has to be a multiple of 3 because of the 3 part. Alright, so the number must end with 0, and the rest of the digits can only be 2s or 0s. Let me think about how to approach this. Maybe I can start by considering numbers made up of 2s and 0s that end with 0 and check if their digit sums are divisible by 3. Let's start with the smallest possible numbers. The smallest multiple of 30 is 30 itself. Is 30 made up only of 0s and 2s? Well, it has a 3 and a 0, so no. So 30 is out. Next, let's try 60. 60 has a 6 and a 0. Again, not just 2s and 0s. So 60 is also out. How about 90? That's 9 and 0. Still not good. Hmm, maybe I need to think differently. Instead of looking at multiples of 30, maybe I should construct numbers with only 2s and 0s that are divisible by 30. Since the number must end with 0, let's consider numbers like 20, 220, 2220, etc. Let's check their divisibility by 3. Starting with 20: The sum of the digits is 2 + 0 = 2. 2 is not divisible by 3, so 20 isn't a multiple of 30. Next, 220: 
...... 
Therefore, 2220 is the smallest such number. Let me double-check: 2220 divided by 30 is 74, which is an integer. So yes, 2220 is a multiple of 30, and it's made up only of 2s and 0s. I think that's the answer. So, the least positive integer multiple of 30 that can be written with only the digits 0 and 2 is 2220.</think> <answer>The least positive integer multiple of 30 that can be written with only the digits 0 and 2 is \boxed{2220}. </answer>**(1555 Tokens)** |

Figure 4: An case study comparing the reasoning process of DeepSeek-R1-Distill-Qwen-1.5B trained with GRPO and ACPO in MATH 500 Dataset.

quickly converging to the correct answer in just 476 tokens. Notably, the model exhibits effective system switch, employing slow thinking during problem analysis and fast thinking for verification once a solution is found, achieving substantial token savings especially in the verification stage by eliminating repetitive constraint checks. The system-aware reasoning tokens offer a transparent view of the model's cognitive process, enabling interpretable analysis of how reasoning behavior aligns with fast and slow thinking modes.

## 5 Related Work

**Efficient Reasoning in LRMs.** Large reasoning models (LRMs) like DeepSeek-R1 [2] and OpenAI o1 [4] have demonstrated strong reasoning capabilities, but their long reasoning paths often suffer from overthinking, introducing redundant content and reducing inference efficiency [10, 13, 11]. To address this, one line of work focuses on supervised fine-tuning with concise, high-quality data to reduce the length of reasoning paths [14, 15]. For instance, TokenSkip [14] generates short reasoning chains through token importance sampling, while Self-Training [15] selects correct short reasoning paths through sampling. Another line of work seeks to improve efficiency through reinforcement learning by modifying reward functions to penalize excessively long reasoning paths [3, 17]. Among these methods, length budgeting is a direct way to explicitly control reasoning length. For example, L1 [18] introduces a fixed length budget to penalize responses that exceed a predefined length during RL training. DAST [19] estimates task difficulty and token length budget through sampling, and constructs preference data for SimPO [32] training. In this work, we integrate online difficulty-aware length budgeting into reinforcement learning process, enabling real-time reasoning budget estimation and reward allocation for efficient reasoning.

**Fast and Slow Thinking in LLMs.** The dual process theory [20] describes two modes of human thinking: fast, intuitive thinking (System 1) and slow, deliberate thinking (System 2). Recent studies have investigated fast and slow thinking in LLMs, focusing on system switch based on task complexity or uncertainty. Specifically, System1.x [34] adapts a controller and System-1/2 planner to adjust reasoning systems based on task difficulty. FaST [35] develops a switching adapter that transitions between System 1 and System 2 for visual reasoning, depending on factors like uncertainty. HaluSearch [36] leverages model performance to generate supervised labels, enabling hierarchical dynamic switch between reasoning systems within MCTS. Dyna-Think[37] implements a training-free dynamic thinking mechanism, allowing the model to autonomously determine when to apply slow reasoning. In this work, we introduce system-aware reasoning tokens to explicitly

represent fast and slow thinking modes, and leverage ACPO with online token length budget to enable adaptive cognitive allocation and dynamic system switch based on task difficulty.

## 6 Conclusion

In this paper, we proposed ACPO, a reinforcement learning framework designed to address the overthinking problem in LRMs through adaptive cognitive allocation and dynamic system switch. In our approach, we first introduce system-aware reasoning tokens to explicitly represent fast and slow thinking modes, making the model's cognitive process transparent and interpretable. Next, we proposed a two-stage training strategy, first fine-tuning the model to establish the ability to generate explicit thinking process, followed by reinforcement learning with ACPO to enhance adaptive cognition allocation. ACPO integrates online difficulty estimation and token length budget during training, guiding dynamic reasoning through a carefully designed reward function. Experimental results demonstrate that ACPO effectively reduces redundant reasoning without sacrificing too much accuracy. Our work provides a flexible and interpretable framework for adaptive hybrid reasoning, supporting efficient and difficulty-aware cognitive processes in LRMs.

## 7 Limitation

Although ACPO effectively reduces redundant reasoning and enables adaptive cognitive allocation, it has several limitations. First, its online difficulty estimation and token length budget mechanisms rely on verifiable data, which may limit generalization to open-domain tasks. Future work should explore more generalizable estimation methods for broader applicability. Second, ACPO can introduce accuracy trade-offs despite improving reasoning efficiency, highlighting the need for more refined reward designs to better balance efficiency and correctness.

## Acknowledgments

This work was partially supported by National Natural Science Foundation of China under Grant No. 92470205 and 62222215, Beijing Natural Science Foundation under Grant No. L233008 and Beijing Municipal Science and Technology Project under Grant No. Z231100010323009. Xin Zhao is the corresponding author.

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

# A   Prompts for Explicit Thinking Annotation

In Section 3.1.1, we prompt DeepSeek-R1-Distill-Qwen-32B to generate responses of varying reasoning lengths for candidate response sampling, and leverage GPT-4 for fine-grained comparison to annotate different thinking modes. The system prompts for these two components are provided below, where the system prompts for candidate sampling are adapted from TOPS [24].

---

**System Prompts for Candidate Response Sampling**

**Low Reasoning Effort:** You have extremely limited time to think and respond to the user's query. Every additional second of processing and reasoning incurs a significant resource cost, which could affect efficiency and effectiveness. Your task is to prioritize speed without sacrificing essential clarity or accuracy. Provide the most direct and concise answer possible. Avoid unnecessary steps, reflections, verification, or refinements UNLESS ABSOLUTELY NECESSARY. Your primary goal is to deliver a quick, clear and correct response.

**High Reasoning Effort:** You have unlimited time to think and respond to the user's question. There is no need to worry about reasoning time or associated costs. Your only goal is to arrive at a reliable, correct final answer. Feel free to explore the problem from multiple angles, and try various methods in your reasoning. This includes reflecting on reasoning by trying different approaches, verifying steps from different aspects, and rethinking your conclusions as needed. You are encouraged to take the time to analyze the problem thoroughly, reflect on your reasoning promptly and test all possible solutions. Only after a deep, comprehensive thought process should you provide the final answer, ensuring it is correct and well-supported by your reasoning.

---

**System Prompts for Response Comparison and Annotation**

Given a problem, a short answer, and a long answer, compare the short answer with the long answer and annotate the short answer based on the following rules. If the short answer omits certain reasoning or calculation steps that are present in the long answer, these omitted steps are considered fast thinking, and the corresponding parts in the short answer should be enclosed within <fast_think></fast_think>. If the short answer contains the same reasoning or calculation steps as the long answer, these parts are considered slow thinking and should be enclosed within <slow_think></slow_think>. Fast thinking parts typically involve intuitive judgments, skipped steps, or direct conclusions, whereas slow thinking parts involve full reasoning or calculations that align with those in the long answer. The output should be the short answer with the appropriate <fast_think></fast_think> and <slow_think></slow_think> tags added.

#Problem#:
#Long Answer#:
#Short Answer#:
#Annotated Answe#:

---

# B   Experimental Setting

## B.1   Training and Evaluation Details

In the SFT stage, the learning rate is $1 \times 10^{-5}$, the batch size is $8$, the number of epochs is $3$. For evaluation, we use Qwen2.5 [38] tokenizer to calculate the number of tokens in the responses generated by each model for a fair comparison. All the experiments are conducted on 16 NVIDIA A100 GPUs.

## B.2   Baseline Details

We present descriptions of the three baseline methods *SFT_Shortest*, *SimPO_Shortest*, and *SimPO_DAST* from DAST in Table 1. We adopt their evaluation results as reported in the original DAST paper [19].

- **SFT_Shortest**: Supervised fine-tuning using only the shortest correct sampled response of each problem as training data.

- **SimPO_Shortest**: SimPO with contrastive instance pairs, which take the shortest correct sampled response of each problem as positive instance and the longest as negative instance.

- **SimPO_DAST**: SimPO with contrastive instance pairs from $D_{pre}$ constructed in DAST.

## B.3   Prompt for ACPO Training

We present the prompts used in the ACPO training process below.

> **Prompt for ACPO Training**
>
> You are a helpful AI Assistant that provides well-reasoned and detailed responses. You first think about the reasoning process as an internal monologue and then provide the user with the answer. Respond in the following format: <think>...</think><answer>...</answer>. In the reasoning process, you think fast in simple steps and slowly in complex steps, and put the slow thinking content in <slow_think></slow_think> and fast thinking content in <fast_think></fast_think>. Let's think step by step and output the final answer within boxed{}.

# C    License for existing assets

We list the licenses for the datasets and models used in our experiments.

## C.1    Models

- DeepSeek-R1-Distill-Qwen-1.5B: Apache 2.0 License.
- DeepSeek-R1-Distill-Qwen-7B: Apache 2.0 License.
- DeepSeek-R1-Distill-Llama-8B: Llama3.1 License.

## C.2    Datasets

- The GSM8K [29] dataset: MIT License.
- The MATH [30] dataset: MIT License.
- The DeepScaleR-Preview-Dataset [26]: MIT License.

