# OpenReview forum: "Incentivizing Dual Process Thinking for Efficient Large Language Model Reasoning"
_NeurIPS.cc/2025/Conference — NeurIPS 2025 poster_

### Official Review · Reviewer_Swif · 2025-07-01

**Clarity:** 3
**Significance:** 2
**Originality:** 3
**Rating:** 4
**Confidence:** 3

**Summary:**

This paper proposes Adaptive Cognition Policy Optimization (ACPO), a reinforcement learning framework that enables large language models to dynamically switch between fast and slow reasoning modes using system-aware tokens. ACPO introduces an online difficulty-aware token length budget and combines it with a multi-part reward function to optimize reasoning efficiency. The two-stage training pipeline includes supervised fine-tuning followed by reinforcement learning with GRPO. Experiments on math benchmarks show improved token efficiency with competitive accuracy. The method aligns reasoning behavior with problem complexity, aiming to mitigate overthinking in LRMs.

**Questions:**

- The performance of the proposed method is worse than L1-max in MATH 500 and AIME 2024 datasets on AUC. Could the author explain why that happens?
- How does the proposed reward function explicitly encourage longer slow thinking segments for more complex tasks during reinforcement learning, beyond what is learned during supervised fine-tuning? Could the authors show the distribution of the fast and slow thing token numbers under different question difficulties?
- Can the authors provide ablation studies to demonstrate the individual contributions of each reward component and address potential reward hacking from the weighted combination? E.g., reward loss figures through training.

**Ethical Concerns:**

["NO or VERY MINOR ethics concerns only"]

**Final Justification:**

Thanks for the response from the authors. My concerns are addressed. I vote for borderline accept now.

**Limitations:**

Yes.

**Quality:**

3

**Strengths And Weaknesses:**

Strengths:
- The dual-process reasoning formulation (fast thinking vs. slow thinking) provides an interpretable and cognitively grounded approach to reasoning in LLMs.
- Results demonstrate a substantial reduction in token usage without significant accuracy loss, with improvements in ACU.

Weakness:
- There is no ablation study to prove the effectiveness of the proposed reward function. Using a weighted sum of multiple reward functions may easily lead to reward hacking.
- The optimization objective is somewhat underspecified. While the SFT stage enforces distinct fast and slow thinking patterns, the RL phase does not directly reinforce the intended structural relationship (e.g., slow thinking should be longer than fast thinking), potentially weakening the cognitive alignment.
- The proposed pipeline is only evaluated on two math datasets without any other task, e.g., code, reasoning gym agent, etc.
- The performance of the proposed method is worse than L1-max in MATH 500 and AIME 2024 datasets on AUC. Could the author explain why that happens?

---

> ### Author Rebuttal · Authors · 2025-07-30
>
> Thanks for your valuable comments. Our responses to your concerns are as follows:
>
> **Q1&W4: Performance of L1-Max**
>
> We would like to clarify that both L1-Exact and L1-Max are trained on DeepScaleR-1.5B-Preview, which is a stronger model obtained by further tuning DeepSeek-R1-Distill-Qwen-1.5B using GRPO. As such, the performance of L1-Max benefits from additional training and a more capable backbone.
>
> In contrast, our ACPO model are trained based on DeepSeek-R1-Distill-Qwen-1.5B, without the additional GRPO training step. Therefore, the comparison is not directly fair in terms of model capacity.
>
> To reflect this, we report L1 baselines separately in Table 1 as reference points, rather than treating them as primary baselines.
>
> **Q2: Reward Function effects on fast and slow thinking behavior**
>
> In the reward computation phase of ACPO, for queries whose estimated difficulty exceeds a predefined threshold, our system pattern reward includes a positive term proportional to the ratio of &lt;slow_think&gt; tokens in the response. This design explicitly encourages the model to allocate more slow thinking to harder questions during reinforcement learning. Conversely, for easier queries (below the threshold), the reward encourages a higher proportion of fast thinking.
>
> As shown in Figure 3, we visualize the distribution of fast vs. slow thinking tokens across different difficulty levels in the MATH dataset. The results show a clear increase in the proportion of slow thinking as task difficulty increases, validating the intended behavior of our reward function.
>
> **Q3&W1: Ablation studies of reward**
>
> To assess the individual contributions of each reward component, we conduct ablation studies by incrementally adding each reward term during training. The results on MATH 500 and AIME 2024 are summarized below.
>
> | Model | MATH 500|| |AIME 2024|| |
> |------------------------------|--------|--------|--------|--------|--------|--------|
> |               | Accuracy   | #Tokens | ACU| Accuracy | #Tokens | ACU     |
> | DeepSeek-R1-Distill-Qwen-1.5B| 83.9 | 5708 |  0.98  | 28.9 | 16894  | 0.11 |
> | +Accuracy Reward       | 84.5 | 3098  | 1.82   |  29.0 |  12990 |  0.15 |
> | + Accuracy + Online TLB Reward |80.7 |1635 | 3.29| 28.9| 6390| 0.30 |
> | + Accuracy + TLB + System Reward | 81.0|1679 |3.22 | 30.0| 6670 | 0.30 |
>
> From the results we can see that:
> 1) The accuracy reward improves task performance while moderately reducing token usage. 2) Adding the online TLB reward significantly compresses reasoning length, though it may slightly affect accuracy due to more aggressive compression. 3) The system pattern reward helps recover some accuracy while maintaining efficient token usage.
>
> We will include reward loss curves and additional analysis in the final version to provide a more comprehensive ablation study.
>
> **W2: The optimization objective of RL phase**
>
> While the SFT stage introduces token-level thinking modes, the RL phase reinforces the intended structural relationship through the system pattern reward. Specifically, this reward is positively correlated with the proportion of &lt;slow_think&gt; or &lt;fast_think&gt; tokens, depending on the estimated difficulty of the question.
>
> For harder questions (above a predefined difficulty threshold), the model receives higher reward for allocating more &lt;slow_think&gt; tokens, promoting more deliberate and thorough reasoning. Conversely, for easier questions, the reward encourages a higher proportion of &lt;fast_think&gt; tokens to favor brevity and efficiency. This dynamic reward design guides the model to allocate cognitive effort adaptively based on task complexity.
>
> As shown in Figure 3, the model learns to produce longer slow thinking segments for more difficult problems, and shorter reasoning traces for easier ones.
>
> **W3: Generalization to other domains**
>
> We understand the reviewer’s concern regarding generalization. While our initial experiments focused on math reasoning, we additionally evaluated our method on the GPQA Diamond dataset. As shown in the table below, ACPO significantly reduces reasoning token usage while maintaining comparable accuracy, demonstrating its potential to generalize beyond math.
>
> | Models | Accuracy|#Token|ACU |
> |----------|----------|----------|----------|
> | DeepSeek-R1-Distill-Qwen-1.5B | 33.8 | 22557 | 0.09 |
> |ACPO-1.5B | 32.3 | 11226| 0.19 |
>
> Our method is task-agnostic and can be extended to other domains involving complex reasoning. Our current training data focuses primarily on math problems. Tasks like code generation and agent pose additional challenges due to the task constraints. Nevertheless, we plan to synthesize fast/slow thinking data in these domains to enable supervised fine-tuning and ACPO training, further enhancing the method’s applicability to a broader range of complex reasoning tasks.

---

> > ### Author Response · Authors · 2025-08-06
> > **Kindly Reminder**
> >
> > Hi Reviewer,
> >
> > Can I please know whether our response addresses your questions? If you have any other questions, we are very happy to discuss with you to address all your concerns.

---

> > > ### Author Response · Authors · 2025-08-08
> > > **Kindly Reminder**
> > >
> > > Dear Reviewer Swif,
> > >
> > > There is very little time left before the reviewer–author discussion stage ends. We still do not know if you have received our response. Could you kindly let us know if our reply addresses your questions? If you have any further concerns, we would be more than happy to discuss them with you to ensure all your points are fully addressed.

---

### Official Review · Reviewer_mRGf · 2025-07-01

**Clarity:** 3
**Significance:** 3
**Originality:** 3
**Rating:** 4
**Confidence:** 4

**Summary:**

The authors of this paper find that Large Reasoning Models (LRMs) tend to "overthink," producing redundant reasoning content regardless of task difficulty. This behavior leads to inefficiency on simple problems and strategic inadequacy on complex ones, where a fixed approach may fail. The root cause, according to the paper, is the models' lack of a mechanism to dynamically adjust their reasoning policy in response to task complexity. To tackle this challenge, the authors introduce the Adaptive Cognition Policy Optimization (ACPO) framework.

**Questions:**

In my view, a major significance of this research lies in its potential for saving computational costs during inference. Could the authors provide more analysis to quantify this advantage? For instance, it would be highly compelling to show that your method achieves comparable performance to previous approaches while using only a fraction (e.g., x%) of the reasoning tokens. This is particularly relevant as redundancy in the reasoning process is a core problem for current LRMs, which your method aims to solve.

**Ethical Concerns:**

["NO or VERY MINOR ethics concerns only"]

**Limitations:**

yes

**Quality:**

4

**Strengths And Weaknesses:**

Strengths
1. The online Token Length Budget (TLB) reward is a well-designed mechanism. Its ability to adaptively adjust the reasoning length based on sample success rates is a notable contribution.

2. The proposed method appears to have significant practical value, especially given the current landscape of large-scale deployment of Large Reasoning Models (LRMs).

Weakness
1. The generalizability of the proposed method is not well-established. The training dataset is based on LIMO, a high-quality dataset for complex mathematical reasoning, and the experiments are exclusively confined to this domain. The paper does not evaluate the approach on other types of reasoning tasks, such as commonsense reasoning. Consequently, it remains uncertain whether the method is effective in scenarios beyond mathematical problem-solving.

---

> ### Author Rebuttal · Authors · 2025-07-30
>
> Thanks for your valuable comments. Our responses to your concerns are as follows:
>
> **W1: Generalization of ACPO**
>
> We understand the reviewer's concern regarding generalization. While our original experiments focused on math reasoning tasks, we agree that broader evaluation is important. To this end, we conducted additional experiments on GPQA Diamond. As shown in the table below, ACPO reduces reasoning token usage while maintaining comparable accuracy, demonstrating its generalization potential beyond math.
>
> | Models | Accuracy|#Token|ACU |
> |----------|----------|----------|----------|
> | DeepSeek-R1-Distill-Qwen-1.5B | 33.8 | 22557 | 0.09 |
> |ACPO-1.5B | 32.3 | 11226| 0.19 |
>
> Our method is task-agnostic and can be extended to other domains involving complex reasoning. In future work, we plan to synthesize more fast/slow thinking data in general domains for supervised fine-tuning and ACPO training, to further enhance its applicability across a wider range of tasks.
>
> **Q1: Quantify the computational saving**
>
> We thank the reviewer for highlighting this important aspect. One of the key motivations of our method is indeed to reduce overthinking and improve reasoning efficiency during inference.
>
> In Table 1 and Table 2, we report Accuracy, Token Count, and ACU to demonstrate the effectiveness of our approach. Additionally, we compute the token compression ratio, defined as:
> Compression Ratio = 1 − tokens(ACPO) / tokens(Baseline),
> which quantifies how much token usage is reduced compared to baseline models.
>
> As shown in the table below, ACPO consistently reduces the number of reasoning tokens across all datasets and backbones. Notably, the maximum compression of ACPO-1.5B reaches 70.58% on the MATH dataset. We will include additional quantitative analyses of token savings in the final version to better illustrate the computational efficiency gains enabled by ACPO.
>
> | Models | GSM8K|MATH 500|AIME 2024|
> |----------|----------|----------|----------|
> |ACPO-1.5B  | 11.04% | 70.58% | 60.52%|
> |ACPO-7B  | 7.19% | 64.67%| 65.89%|
> |ACPO-8B  | 28.65%| 55.38%| 54.77% |

---

### Official Review · Reviewer_wub4 · 2025-07-02

**Clarity:** 4
**Significance:** 3
**Originality:** 3
**Rating:** 5
**Confidence:** 5

**Summary:**

This paper aims to train the LLM to think (or to reason) in two speeds, specifically fast thinking and slow thinking. Here, the fast thinking is for system 1 for easy reasoning problems (or easy reasoning stage, such as verification after answering), while slow thinking is for complex reasoning. To achieve this, the author proposes a model to think by using special tokens, namely \<fast_thinking\> and \<slow_thinking\>. To train this model, the author proposes SFT as a cold start by collecting slow and fast thinking processes, then uses online RL with a new reward that forces the model to think efficiently.

**Questions:**

See the weakness above

**Ethical Concerns:**

["NO or VERY MINOR ethics concerns only"]

**Final Justification:**

The rebuttal has well addressed my concern, thus increasing my score from 4 (to 5).

**Limitations:**

See the weakness above

**Paper Formatting Concerns:**

No concerns regarding paper formatting

**Quality:**

3

**Strengths And Weaknesses:**

**Strengths**

(1) I think the problem is well-motivated, and the method is quite novel. I do like the special tokens, \<fast_think\> and \<slow_think\>. Also, having a cold start with SFT makes sense.

(2) Overall, the writing is clear

(3) The experiments are also well conducted.

I do not find major weaknesses in this paper, but I do have some concerns that would be great if the authors could resolve them through the rebuttal.

**Weakness**

(1) I think the paper has only focused on the math domain, which is one limitation. I think it would be great if the authors could extend to coding (e.g., MBPP, Humaneval) or general domains (e.g., GPQA diamond).

(2) Additionally, I believe it is worthwhile to analyze the idea on more challenging benchmarks. It is great that the authors have already considered AIME 2024, but also AIME 2025, and AMC 2023 would be worth it. The motivation behind this question is that slow thinking may appear more in challenging scenarios compared to easy ones (e.g., GSM 8K).

(3) I quite like the idea of SFT warmup (i.e., Cold start). However, I could not find the effect of this cold start in the paper. Can the authors provide some analysis behind this?

(4) This is not a weakness rather a question. Can the author provide the intuition behind the reward design? For instance, using negative reward for a wrong answer is not common. Is there any intuition behind this?

(5) I think adding GRPO as a default baseline for all datasets will be required (i.e., in Table 2 the GRPO is missing). Also, when adding more datasets during the rebuttal, I kindly ask the authors to add GRPO as a baseline.

Overall, I think the paper is well written and has the strength to be accepted. I kindly ask the authors to address the minor concerns that I have (I am willing to increase the score after the rebuttal).

---

> ### Author Rebuttal · Authors · 2025-07-30
>
> Thanks for your valuable comments. Our responses to your concerns are as follows:
>
> **Q1: Generalization to other domains**
>
> We understand the reviewer’s concern regarding generalization. While our initial experiments focused on math reasoning, we additionally evaluated our method on the GPQA Diamond dataset. As shown in the table below, ACPO significantly reduces reasoning token usage while maintaining comparable accuracy, demonstrating its potential to generalize beyond math.
>
> | Models | Accuracy|#Token|ACU |
> |----------|----------|----------|----------|
> | DeepSeek-R1-Distill-Qwen-1.5B | 33.8 | 22557 | 0.09 |
> |ACPO-1.5B | 32.3 | 11226| 0.19 |
>
> Our method is task-agnostic and can be extended to other domains involving complex reasoning. Our current training data focuses primarily on math problems. Tasks like code generation pose additional challenges due to some formatting constraints. Nevertheless, we plan to synthesize fast/slow thinking data in these domains to enable supervised fine-tuning and ACPO training, further enhancing the method’s applicability to a broader range of complex reasoning tasks.
>
> **Q2: Evaluation on more challenging benchmarks**
>
> To strengthen our evaluation on challenging benchmarks, we tested our method on AIME 2025 and AMC 2023. As shown below, ACPO achieves effective token compression on both datasets. Notably, it achieves higher compression on the simpler AMC 2023, and on the more difficult AIME 2025, it improves accuracy while reducing reasoning length, further demonstrating the effectiveness of our method in adapting reasoning length to task complexity.
>
> | Model                          |        AMC 2023         |                    |        |        AIME 2025        |                    |        |
> |-------------------------------|--------------------------|--------------------|--------|--------------------------|--------------------|--------|
> |                               | Accuracy                 | #Tokens            | ACU    | Accuracy                 | #Tokens            | ACU    |
> | DeepSeek-R1-Distill-Qwen-1.5B | 70.3                     | 10030              | 0.46   | 25.0                     | 18431              | 0.09   |
> | ACPO-1.5B                     | 69.2                     | 3270               | 1.41   | 30.0                     | 10690              | 0.19   |
>
> **Q3：Effect of SFT warmup**
>
> We thank the reviewer for highlighting the cold-start phase. Its main purpose is not to boost accuracy, but to teach the model to emit &lt;fast_think&gt; and &lt;slow_think&gt; tokens in a consistent and semantically meaningful way. This structured initialization lays the foundation for the reinforcement learning phase.
>
> To assess the impact of this phase, we report the model's accuracy before and after SFT. As shown below, the accuracy remains largely stable. We will clarify this point more explicitly in the future version.
>
> | Models | GSM8K | MATH 500 | AIME 2024|
> |----------|----------|----------|----------|
> |DeepSeek-R1-Distill-Qwen-1.5B  | 79.9|83.9|28.9|
> | +SFT | 80.4 | 83.0| 26.7|
> |DeepSeek-R1-Distill-Qwen-7B  | 86.5 | 92.8 | 55.5|
> | +SFT | 87.4 | 90.3| 53.3|
> |DeepSeek-R1-Distill-Llama-8B  | 82.9 | 89.1| 42.9|
> | +SFT |87.2 | 88.2| 42.0|
>
> **Q4: Intuition behind reward design**
>
> For tasks like math problem solving, answer correctness is the primary objective, and any reasoning trace—no matter how efficient or well-structured—is ultimately invalid if it leads to the wrong answer.
>
> To enforce this principle, we assign a mild negative reward to incorrect answers, ensuring that no positive reward is given unless the final prediction is correct. Using zero reward for incorrect answers can unintentionally result in a positive total reward if the reasoning trace scores well on length or structure, which contradicts the task goal.
>
> This design encourages the model to first guarantee correctness, and then optimize for reasoning efficiency and cognitive control. We found this constraint important to prevent reward hacking and guide the model toward meaningful improvements.
>
> **Q5：GRPO baseline results**
>
> We thank the reviewer for pointing this out. We will supplement Table 2 with the corresponding GRPO results and include GRPO as a baseline across all evaluated datasets in the updated version.
>
> | Models | Accuracy|#Token|ACU |
> |----------|----------|----------|----------|
> |GRPO-1.5B | 81.5 | 621| 8.72 |
> |GRPO-7B | 88.3 | 429| 2.94 |
> |GRPO-8B | 87.9 | 857| 1.28 |

---

> > ### Comment · Reviewer_wub4 · 2025-08-05
> > **Thank you for the rebuttal**
> >
> > Thank you for taking the time to address my concerns. I believe the additional experiments have strengthened the paper's claim and make it worth being accepted to NeurIPS. I raised my score accordingly.

---

> > > ### Author Response · Authors · 2025-08-05
> > > **Thank You for Your Feedback**
> > >
> > > Thank you for revisiting our paper and for your encouraging feedback. We're really glad to hear that the additional experiments helped clarify our contributions. We appreciate your updated score and support.

---

### Official Review · Reviewer_dgDW · 2025-07-05

**Clarity:** 3
**Significance:** 4
**Originality:** 3
**Rating:** 5
**Confidence:** 3

**Summary:**

This paper introduces a novel method to teach large reasoning models (LRMs) how to incorporate 'fast' and 'slow' thinking adaptively while solving mathematical reasoning problems. A small dataset is curated using rejection sampling and LLM-as-judge to annotate correct, shorter (longer) responses as fast (slow) thinking examples. Subsequently, a language model is supervised fine-tuned for cold start to generate chain-of-thought with fast and slow thinking token markers. The ensuing RL training is primarily based on GRPO but includes two additional rewards: online token length budget that calculates the token budget adaptively from the GRPO sampled candidate answers and their success rate, and an additional reward to incentivize slow thinking for harder problems and fast thinking for easier problems (sample success rate is used a proxy for task difficulty). Three testbeds are used to evaluate the effectiveness of the proposed method.

**Questions:**

1. How do you interleave two modes in the response? Figure 1 tells that the mode annotation is per-instance and not per-token.

2. Did you perform manual checking of the GPT generated annotation? Given the quality of the LIMO dataset, I doubt GPT-4 will be able to correctly identify all cases of reasoning nuances. What happens if the short and long responses are explicitly two different state-transition sequences?

3. How do you differentiate between short and long (fast or slow) reasoning? Can you provide the length-wise distribution of the responses and where you put the decision boundary?

3. The LIMO paper itself used the same testbeds and fewer training data. The accuracy numbers are much higher than all the numbers in this paper. I would like to have your comment on this.

4. The dual-mode thinking is a debated hypothesis in cognitive science (see Strengths and Weaknesses). I would suggest the authors take into account and correspondingly position their contribution and limitations.

5. How does the token budget affect the performance? Can you analyse cases where the baseline model (that used a longer reasoning trace) was able to solve the problem, but ACPO failed?

6. Analysis of ACPO on distribution shift (e.g., GSM-plus [1]) compared to a standard (overthinking) model will significantly improve the quality of the paper.

[1] Li, Q., et al. (2024). A comprehensive benchmark for evaluating the robustness of LLMs as mathematical problem solvers.

**Ethical Concerns:**

["NO or VERY MINOR ethics concerns only"]

**Final Justification:**

Author's response could address my initial concern.

**Limitations:**

Yes

**Quality:**

4

**Strengths And Weaknesses:**

## Strengths:

- Highly relevant research problem: Most LRMs suffer heavily from overthinking, leading to erroneous reasoning as well as high computational cost.

- Interesting approach to solution: I really like the idea of interleaving slow and fast thinking. Personal opinion: Differentiating between fast or slow thinking in absolute terms of the problem (i.e., this problem requires system 1 thinking or that problem requires system 2 thinking) is a debatable topic in cognitive science (see [1] or [2], for example). The interleaving can be thought of as an alternative to such hard binaries. I would really appreciate if the authors put some thought into this direction to further strengthen the argument of the paper.

- Cheap reasoning trace: Proposed method renders the model to use tokens in an efficient manner. This (along with similar papers in this direction) can provide cheaper effective alternatives.

# Weaknesses

- Occasional lack of clarity: Certain major details are missing (or, I could not find them effectively), for example, how they make the model to interleave two different modes within a single chain-of-thought when the data is per-instance, what happens (in the data curation phase) is the longer vs shorter reasoning traces explore two different paths of reasoning altogether (that leads to the same answer, but uses different ways or premises), why certain experiments are not performed with GSM8K, etc.

- Narrow focus: The proposed method is designed and experimented with mathematical reasoning only, and no pointers are provided for if and how it can be adapted to other reasoning tasks.

- Insufficient analysis of errors: While the limitation section is quite superficial in my opinion, there is no explicit analysis of erroneous cases.

[1] Keren, G., & Schul, Y. (2009). Two is not always better than one: A critical evaluation of dual-system theories.

[2] Melnikoff, D. E., & Bargh, J. A. (2018). The Mythical Number Two.

---

> ### Author Rebuttal · Authors · 2025-07-30
>
> Thanks for your valuable comments. Our responses to your concerns are as follows:
>
> **Q1: Clarification on how fast and slow thinking modes are interleaved**
>
> The Dual Process Reasoning in Figure 1 indeed corresponds to step-level thinking-mode annotations, not per-instance labels. Each response includes interleaved &lt;fast_think&gt; and &lt;slow_think&gt; tags to mark different reasoning steps, enabling fine-grained cognitive switching within a single trace. This interleaving behavior is learned through a two-stage process: supervised fine-tuning with explicitly annotated data and reinforcement learning with system-aware rewards. A concrete token-level example is shown in Figure 4. We will revise Figure 1 to clarify this more explicitly.
>
> **Q2：Annotation data quality**
>
> We acknowledge that annotation quality is critical. In our pipeline, we use the same model with different prompts to generate short and long responses, which encourages differences in reasoning granularity while largely preserving the core solution structure.
>
> We rely on GPT-4 to perform the annotation automatically and filter out any samples where GPT-4 fails to annotate due to structural mismatch or unclear reasoning trace. We spot-checked a subset of samples to verify that the generated tags aligned with our intended labeling protocol.
>
> We appreciate the reviewer’s suggestion and will further review all annotated data to ensure consistency and quality.
>
> **Q3: Differentiation between fast and slow reasoning**
>
> In our framework, the distinction between fast and slow reasoning is established during data construction by prompting the same model with different instructions. The short responses are encouraged to be concise and direct, while the long responses are prompted to be more thorough and elaborative.
>
> To obtain step-level annotations, we compare the short and long response pairs using GPT-4. Reasoning steps shared by both responses are labeled as slow thinking, while steps present only in the long response are labeled as fast thinking. This approach does not rely on a fixed token-length threshold, but instead uses semantic comparison to determine reasoning granularity.
>
> Below, we report the average token lengths of short, long, and annotated responses to provide a reference for their length distribution.
>
> | Response Type | Long Response | Short Response | Annotated Response |
> |----------|----------|----------|----------|
> |Tokens   | 7105 | 2683  | 3020 |
>
> **Q4: Comparison with LIMO performance**
>
> We would like to clarify that the LIMO paper uses Qwen2.5-32B-Instruct as its backbone for supervised fine-tuning, while our experiments are conducted on smaller models (DeepSeek-R1-Distill-Qwen-1.5B, 7B, and DeepSeek-R1-Distill-LLaMA-8B). Given the significant gap in model capacity, the accuracy numbers are not directly comparable.
>
> Additionally, our SFT phase does not use the original LIMO solutions, but the newly generated reasoning paths annotated with &lt;fast_think&gt; and &lt;slow_think&gt; tokens.
>
> To isolate the impact of backbone, we fine-tuned DeepSeek-R1-Distill-Qwen-1.5B on the original LIMO dataset to serve as a fair comparison. As shown in the table below, the performance of the trained model is even lower than our ACPO results.
>
> Finally, we would like to emphasize that our method is not solely aimed at maximizing accuracy, but at achieving comparable performance with more efficient (i.e., shorter) reasoning traces.
>
> | Models | GSM8K | MATH 500 | AIME 2024|
> |----------|----------|----------|----------|
> |DeepSeek-R1-Distill-Qwen-1.5B| 79.9|83.9|28.9|
> |1.5B-limo-sft  | 76.2 | 74.6 | 23.3 |
> | ACPO-1.5B | 81.3 | 81.0 | 30.0 |
>
> **Q5: Controversies of Dual-Mode Thinking**
>
> We are pleasantly surprised and grateful that the reviewer recognized our method as an alternative to the hard binary of fast and slow thinking. In fact, our step-level thinking-mode switching is designed to reflect the dynamic nature of the reasoning process, where fast and slow thinking are interleaved rather than strictly separated.
>
> Rigidly dividing cognition into two distinct modes, or assigning a fixed mode to each problem, is a coarse approach. What inspires us from the dual process theory is not the binary classification itself, but the notion of dynamic cognitive adjustment during reasoning.
>
> We will further clarify this point in the final version to better explain our motivation.
>
> **Q6: Impact of token budget on performance and failure cases**
>
> While ACPO achieves substantial token savings, we acknowledge that aggressive compression may occasionally lead to performance degradation.
>
> We conducted case studies where the baseline model answered correctly but the ACPO-trained model failed. On these cases, baseline responses averaged 7691 tokens, while ACPO responses averaged 3999 tokens.
> The baseline typically explored more intermediate steps and reasoning paths, while ACPO missed certain details due to insufficient exploration. These failures mostly occurred in more challenging cases where longer reasoning is more beneficial. However, our dynamic budget mechanism helps reduce some of these failures by allocating more tokens to harder problems.
>
> Due to space constraints, we will show the detailed case study in the final version to further clarify this point.
>
> **Q7: Analysis of ACPO on distribution shift**
>
> To assess ACPO’s robustness under distribution shift, we evaluated both the baseline and ACPO-trained models on GSM-Plus.
>
> As shown in the table below, ACPO maintains comparable accuracy while effectively reducing the reasoning length, demonstrating its generalization ability under distribution shift.
>
> Models | Accuracy|#Token|ACU |
> |----------|----------|----------|----------|
> | DeepSeek-R1-Distill-Qwen-1.5B | 62.62 |10939 | 0.38 |
> |ACPO-1.5B | 61.25 | 8141| 0.50 |
>
> **W1：Some detailed explanations**
>
> Some experiments were not conducted on GSM8K because it is relatively simple, and the compared baselines did not consider it as a target test set. Additionally, due to the lack of released models or code, or missing reported results, we were unable to include them for comparison. We will include GSM8K results once the corresponding baseline models become available.
>
> **W2: Generalization to other domains**
>
> While our original experiments focused on math reasoning tasks, we agree that broader evaluation is important. To this end, we conducted additional experiments on GPQA Diamond. As shown in the table below, ACPO reduces reasoning token usage while maintaining comparable accuracy, demonstrating its generalization potential beyond math.
>
> | Models | Accuracy|#Token|ACU |
> |----------|----------|----------|----------|
> | DeepSeek-R1-Distill-Qwen-1.5B | 33.8 | 22557 | 0.09 |
> |ACPO-1.5B | 32.3 | 11226| 0.19 |
>
> Our method is task-agnostic and can be extended to other domains involving complex reasoning. In future work, we plan to synthesize more fast/slow thinking data in general domains for supervised fine-tuning and ACPO training, to further enhance its applicability across a wider range of tasks.

---

### Note · Authors · 2025-08-12

We sincerely thank all reviewers and the Area Chairs for their efforts. We are encouraged by the positive recognition from the reviewers:

- **Adaptive dual-process reasoning (Reviewer dgDW, Reviewer wub4)**: The proposed interleaving of fast and slow thinking is novel and well-motivated. Step-level switching offers more flexible and fine-grained control within a single reasoning trace.
- **Dynamic reasoning budget (Reviewer mRGf, Reviewer Swif)**: The Online Token Length Budget is a well-designed mechanism for adaptively adjusting reasoning length based on task difficulty.
- **Efficiency gains (Reviewer dgDW, Reviewer mRGf, Reviewer Swif)**: The method substantially reduces token usage, providing a cheaper, more efficient reasoning process.
- **Clarity (Reviewer wub4)**: The paper is well-written, with a clear presentation of the two-stage pipeline and well-conducted experiments.

**Our responses to reviewer concerns:**
- **Generalization beyond mathematics**: Added experiments on GPQA Diamond, confirming ACPO’s effectiveness in non-math domains, achieving substantial token savings while maintaining comparable accuracy.
- **Evaluation on more challenging benchmarks**: Included results on AIME 2025 and AMC 2023, demonstrating ACPO’s ability to adapt reasoning length dynamically to varying task complexities.
- **Clarity on interleaving fast and slow thinking & role of the SFT cold-start**: Highlighted that mode annotation is step-level, enabling fine-grained cognitive switching within a single reasoning trace. Clarified that the SFT phase’s primary role is to train the model to produce the defined reasoning tokens, laying the groundwork for RL training.
- **Reward design and ablation**: Detailed the design of each RL reward component, showing how it promotes both cost control and dynamic system switch. Provided ablation studies confirming each component’s contribution.
- **Performance comparison with L1**: Clarified that the L1 baseline is trained on a stronger backbone with additional GRPO fine-tuning. As detailed in our experimental section, these differences explain the performance gap.

We appreciate that Reviewer wub4 updated the score after reviewing our rebuttal, and we thank all reviewers who actively participated in the discussion stage. We believe the detailed responses, new analyses, and supplementary experiments we have provided can address the remaining concerns. We will incorporate these key points and clarifications into the revised version.

---

### Decision · Program_Chairs · 2025-09-17

**Decision:**

Accept (poster)

**Comment:**

The domain of experiments limited but this is a nice idea and after the author response was a consensus accept.